# Robust Low Rank Kernel Embeddings of Multivariate Distributions

**Le Song, Bo Dai**
College of Computing, Georgia Institute of Technology
`lsong@cc.gatech.edu, bodai@gatech.edu`

## Abstract

Kernel embedding of distributions has led to many recent advances in machine learning. However, latent and low rank structures prevalent in real world distributions have rarely been taken into account in this setting. Furthermore, no prior work in kernel embedding literature has addressed the issue of robust embedding when the latent and low rank information are misspecified. In this paper, we propose a hierarchical low rank decomposition of kernels embeddings which can exploit such low rank structures in data while being robust to model misspecification. We also illustrate with empirical evidence that the estimated low rank embeddings lead to improved performance in density estimation.

## 1 Introduction

Many applications of machine learning, ranging from computer vision to computational biology, require the analysis of large volumes of high-dimensional continuous-valued measurements. Complex statistical features are commonplace, including multi-modality, skewness, and rich dependency structures. Kernel embedding of distributions is an effective framework to address challenging problems in this regime [1, 2]. Its key idea is to implicitly map distributions into potentially infinite dimensional feature spaces using kernels, such that subsequent comparison and manipulation of these distributions can be achieved via feature space operations (*e.g.*, inner product, distance, projection and spectral analysis). This new framework has led to many recent advances in machine learning such as kernel independence test [3] and kernel belief propagation [4].

However, algorithms designed with kernel embeddings have rarely taken into account latent and low rank structures prevalent in high dimensional data arising from various applications such as gene expression analysis. While these information have been extensively exploited in other learning contexts such as graphical models and collaborative filtering, their use in kernel embeddings remains scarce and challenging. Intuitively, these intrinsically low dimensional structures of the data should reduce the effect number of parameters in kernel embeddings, and allow us to obtain a better estimator when facing with high dimensional problems.

As a demonstration of the above intuition, we illustrate the behavior of low rank kernel embeddings (which we will explain later in more details) when applied to density estimation (Figure 1). 100 data points are sampled *i.i.d.* from a mixture of 2 spherical Gaussians, where the latent variable is the cluster indicator. The fitted density based on an ordinary kernel density estimator has quite different contours from the ground truth (Figure 1(b)), while those provided by low rank embeddings appear to be much closer to the ground truth ((Figure 1(c)). Essentially, the low rank approximation step endows kernel embeddings with an additional mechanism to smooth the estimator which can be beneficial when the number of data points is small and there are clusters in the data. In our later more systematic experiments, we show that low rank embeddings can lead to density estimators which can significantly improve over alternative approaches in terms of held-out likelihood.

While there are a handful of exceptions [5, 6] in the kernel embedding literature which have exploited latent and low rank information, these algorithms are not robust in the sense that, when such information are misspecification, no performance guarantee can be provided and these algorithms can fail drastically. The hierarchical low rank kernel embeddings we proposed in this paper can be

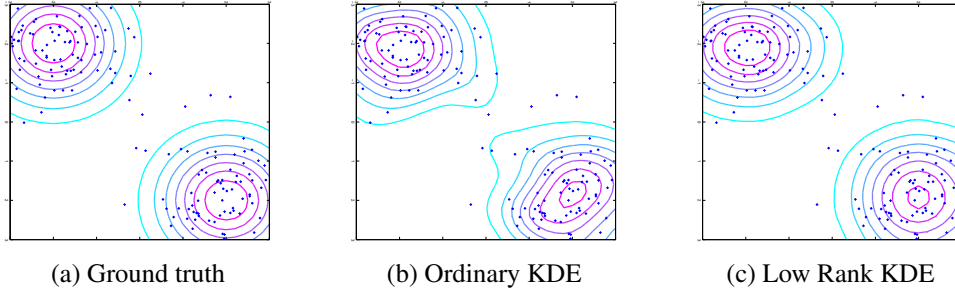

| (a) Ground truth | (b) Ordinary KDE | (c) Low Rank KDE |

Figure 1: We draw 100 samples from a mixture of 2 spherical Gaussians with equal mixing weights. (a) the contour plot for the ground truth density, (b) for ordinary kernel density estimator (KDE), (c) for low rank KDE. We used cross-validation to find the best kernel bandwidth for both the KDE and low rank KDE. The latter produces a density which is visibly closer to the ground truth, and in term of the integrated square error, it is smaller than the KDE (0.0092 vs. 0.012).

considered as a kernel generalization of the discrete valued tree-structured latent variable models studied in [7]. The objective of the current paper is to address previous limitations of kernel embeddings as applied to graphical models and make them more practically useful. Furthermore, we will provide both theoretical and empirical support to the new approach.

Another key contribution of the paper is a novel view of kernel embedding of multivariate distributions as infinite dimensional higher order tensors, and the low rank structure of these tensors in the presence of latent variables. This novel view allows us to introduce modern multi-linear algebra and tensor decomposition tools to address challenging problems in the interface between kernel methods and latent variable models. We believe our work will play a synergistic role in bridging together largely separate areas in machine learning research, including kernel methods, latent variable models, and tensor data analysis.

In the remainder of the paper, we will first present the tensor view of kernel embeddings of multivariate distributions and its low rank structure in the presence of latent variables. Then we will present our algorithm for hierarchical low rank decomposition of kernel embeddings by making use of a sequence of nested kernel singular value decompositions. Last, we will provide both theoretical and empirical support to our proposed approach.

## 2 Kernel Embeddings of Distributions

We will focus on continuous domains, and denote $X$ a random variable with domain $\Omega$ and density $p(X)$. The instantiations of $X$ are denoted by lower case character, $x$. A reproducing kernel Hilbert space (RKHS) $\mathcal{F}$ on $\Omega$ with a kernel $k(x, x')$ is a Hilbert space of functions $f : \Omega \mapsto \mathbb{R}$ with inner product $\langle \cdot, \cdot \rangle_{\mathcal{F}}$. Its element $k(x, \cdot)$ satisfies the reproducing property: $\langle f(\cdot), k(x, \cdot) \rangle_{\mathcal{F}} = f(x)$, and consequently, $\langle k(x, \cdot), k(x', \cdot) \rangle_{\mathcal{F}} = k(x, x')$, meaning that we can view the evaluation of a function $f$ at any point $x \in \Omega$ as an inner product. Alternatively, $k(x, \cdot)$ can be viewed as an implicit feature map $\phi(x)$ where $k(x, x') = \langle \phi(x), \phi(x') \rangle_{\mathcal{F}}$. For simplicity of notation, we assumes that the domain of all variables are the same and the same kernel function is applied to all variables.

A kernel embedding represents a density by its expected features, *i.e.*, $\mu_X := \mathbb{E}_X [\phi(X)] = \int_{\Omega} \phi(x) p(x) dx$, or a point in a potentially infinite-dimensional and implicit feature space of a kernel [8, 1, 2]. The embedding $\mu_X$ has the property that the expectation of any RKHS function $f \in \mathcal{F}$ can be evaluated as an inner product in $\mathcal{F}$, $\langle \mu_X, f \rangle_{\mathcal{F}} := \mathbb{E}_X[f(X)]$. Kernel embeddings can be readily generalized to joint density of $d$ variables, $X_1, \ldots, X_d$, using $d$-th order tensor product feature space $\mathcal{F}^d$, In this feature space, the feature map is defined as $\otimes_{i=1}^d \phi(x_i) := \phi(x_1) \otimes \phi(x_2) \otimes \ldots \otimes \phi(x_d)$, and the inner product in this space satisfies $\langle \otimes_{i=1}^d \phi(x_i), \otimes_{i=1}^d \phi(x_i') \rangle_{\mathcal{F}^d} = \prod_{i=1}^d \langle \phi(x_i), \phi(x_i') \rangle_{\mathcal{F}} = \prod_{i=1}^d k(x_i, x_i')$. Then we can embed a joint density $p(X_1, \ldots, X_d)$ into a tensor product feature space $\mathcal{F}^d$ by

$$\mathcal{C}_{X_{1:d}} := \mathbb{E}_{X_{1:d}} \left[ \otimes_{i=1}^d \phi(X_i) \right] = \int_{\Omega^d} \left( \otimes_{i=1}^d \phi(x_i) \right) p(x_1, \ldots, x_d) \prod_{i=1}^d dx_i, \qquad (1)$$

where we used $X_{1:d}$ to denote the set of variables $\{X_1, \ldots, X_d\}$.

The kernel embeddings can also be generalized to conditional densities $p(X|z)$ [9]

$$\mu_{X|z} := \mathbb{E}_{X|z}[\phi(X)] = \int_\Omega \phi(x)\, p(x|z)\, dx \qquad (2)$$

Given this embedding, the conditional expectation of a function $f \in \mathcal{F}$ can be computed as $\mathbb{E}_{X|z}[f(X)] = \langle f, \mu_{X|z} \rangle_{\mathcal{F}}$. Unlike the ordinary embeddings, an embedding of conditional distribution is not a single element in the RKHS, but will instead sweep out a family of points in the RKHS, each indexed by a fixed value $z$ of the conditioning variable $Z$. It is only by fixing $Z$ to a particular value $z$, that we will be able to obtain a single RKHS element, $\mu_{X|z} \in \mathcal{F}$. In other words, conditional embedding is an operator, denoted as $\mathcal{C}_{X|Z}$, which can take as input an $z$ and output an embedding, *i.e.*, $\mu_{X|z} = \mathcal{C}_{X|Z}\phi(z)$. Likewise, kernel embedding of conditional distributions can also be generalized to joint distribution of $d$ variables.

We will represent an observation from a discrete variable $Z$ taking $r$ possible value using the standard basis in $R^r$ (or one-of-$r$ representation). That is when $z$ takes the $i$-th value, the $i$-th dimension of vector $z$ is 1 and other dimensions 0. For instance, when $r = 3$, $Z$ can take three possible value $(1,0,0)^\top$, $(0,1,0)^\top$ and $(0,0,1)^\top$. In this case, we let $\phi(Z) = Z$ and use the linear kernel $k(Z,Z') = Z^\top Z$. Then, the conditional embedding operator reduces to a separate embedding $\mu_{X|z}$ for each conditional density $p(X|z)$. Conceptually, we can concatenate these $\mu_{X|z}$ for different value of $z$ in columns $\mathcal{C}_{X|Z} := (\mu_{X|z=(1,0,0)^\top}, \mu_{X|z=(0,1,0)^\top}, \mu_{X|z=(0,0,1)^\top})$. The operation $\mathcal{C}_{X|Z}\phi(z)$ essentially picks up the corresponding embedding (or column).

## 3   Kernel Embeddings as Infinite Dimensional Higher Order Tensors

The above kernel embedding $\mathcal{C}_{X_{1:d}}$ can also be viewed as a multi-linear operator (tensor) of order $d$ mapping from $\mathcal{F} \times \ldots \times \mathcal{F}$ to $\mathbb{R}$. (For generic introduction to tensor and tensor notation, please see [10]). The operator is linear in each argument (mode) when fixing other arguments. Furthermore, the application of the operator to a set of elements $\{f_i \in \mathcal{F}\}_{i=1}^d$ can be defined using the inner product from the tensor product feature space, *i.e.*,

$$\mathcal{C}_{X_{1:d}} \bullet_1 f_1 \bullet_2 \ldots \bullet_d f_d := \left\langle \mathcal{C}_{X_{1:d}},\, \otimes_{i=1}^d f_d \right\rangle_{\mathcal{F}^d} = \mathbb{E}_{X_{1:d}}\left[ \prod_{i=1}^d \langle \phi(X_i), f_i \rangle_{\mathcal{F}} \right], \qquad (3)$$

where $\bullet_i$ means applying $f_i$ to the $i$-th argument of $\mathcal{C}_{X_{1:d}}$. Furthermore, we can define the generalized Frobenius norm $\|\cdot\|_\bullet$ of $\mathcal{C}_{X_{1:d}}$ as $\|\mathcal{C}_{X_{1:d}}\|_\bullet^2 = \sum_{i_1=1}^\infty \cdots \sum_{i_d=1}^\infty (\mathcal{C}_{X_{1:d}} \bullet_1 e_{i_1} \bullet_2 \ldots \bullet_d e_{i_d})^2$ using an orthonormal basis $\{e_i\}_{i=1}^\infty \subset \mathcal{F}$. We can also define the inner product for the space of such operator that $\|\mathcal{C}_{X_{1:d}}\|_\bullet < \infty$ as

$$\left\langle \mathcal{C}_{X_{1:d}}, \widetilde{\mathcal{C}}_{X_{1:d}} \right\rangle_\bullet = \sum_{i_1=1}^\infty \cdots \sum_{i_d=1}^\infty (\mathcal{C}_{X_{1:d}} \bullet_1 e_{i_1} \bullet_2 \ldots \bullet_d e_{i_d})(\widetilde{\mathcal{C}}_{X_{1:d}} \bullet_1 e_{i_1} \bullet_2 \ldots \bullet_d e_{i_d}). \qquad (4)$$

When $\mathcal{C}_{X_{1:d}}$ has the form of $\mathbb{E}_{X_{1:d}}\left[ \otimes_{i=1}^d \phi(X_i) \right]$, the above inner product reduces to $\mathbb{E}_{X_{1:d}}[\widetilde{\mathcal{C}}_{X_{1:d}} \bullet_1 \phi(X_1) \bullet_2 \ldots \bullet_d \phi(X_d)]$.

In this paper, the ordering of the tensor modes is not essential so we simply label them using the corresponding random variables. We can reshape a higher order tensor into a lower order tensor by partitioning its modes into several disjoint groups. For instance, let $\mathscr{I}_1 = \{X_1, \ldots, X_s\}$ be the set of modes corresponding to the first $s$ variables and $\mathscr{I}_2 = \{X_{s+1}, \ldots, X_d\}$. Similarly to the Matlab function, we can obtain a 2nd order tensor by

$$\mathcal{C}_{\mathscr{I}_1;\mathscr{I}_2} = reshape\left(\mathcal{C}_{X_{1:d}}, \mathscr{I}_1, \mathscr{I}_2\right) \ : \ \mathcal{F}^s \times \mathcal{F}^{d-s} \mapsto \mathbb{R}. \qquad (5)$$

In the reverse direction, we can also reshape a lower order tensor into a higher order one by further partitioning certain mode of the tensor. For instance, we can partition $\mathscr{I}_1$ into $\mathscr{I}_1' = \{X_1, \ldots, X_t\}$ and $\mathscr{I}_1'' = \{X_{t+1}, \ldots, X_s\}$, and turn $\mathcal{C}_{\mathscr{I}_1;\mathscr{I}_2}$ into a 3rd order tensor by

$$\mathcal{C}_{\mathscr{I}_1';\mathscr{I}_1'';\mathscr{I}_2} = reshape\left(\mathcal{C}_{\mathscr{I}_1;\mathscr{I}_2}, \mathscr{I}_1', \mathscr{I}_1'', \mathscr{I}_2\right) \ : \ \mathcal{F}^t \times \mathcal{F}^{s-t} \times \mathcal{F}^{d-s} \mapsto \mathbb{R}. \qquad (6)$$

Note that given a orthonormal basis $\{e_i\}_{i=1}^\infty \in \mathcal{F}$, we can readily obtain an orthonormal basis for, *e.g.*, $\mathcal{F}^t$, as $\{e_{i_1} \otimes \ldots \otimes e_{i_t}\}_{i_1,\ldots,i_t=1}^\infty$, and hence define the generalized Frobenius norm for $\mathcal{C}_{\mathscr{I}_1;\mathscr{I}_2}$ and $\mathcal{C}_{\mathscr{I}_1';\mathscr{I}_1'';\mathscr{I}_2}$. This also implies that the generalized Frobenius norms are the same for all these reshaped tensors, *i.e.*, $\|\mathcal{C}_{X_{1:d}}\|_\bullet = \|\mathcal{C}_{\mathscr{I}_1;\mathscr{I}_2}\|_\bullet = \|\mathcal{C}_{\mathscr{I}_1';\mathscr{I}_1'';\mathscr{I}_2}\|_\bullet$.

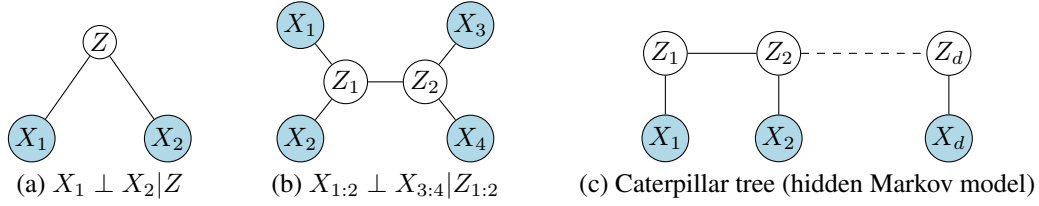

(a) $X_1 \perp X_2 | Z$      (b) $X_{1:2} \perp X_{3:4} | Z_{1:2}$      (c) Caterpillar tree (hidden Markov model)

Figure 2: Three latent variable model with different tree topologies

The 2nd order tensor $\mathcal{C}_{\mathscr{I}_1;\mathscr{I}_2}$ can also be viewed as the cross-covariance operator between two sets of variables in $\mathscr{I}_1$ and $\mathscr{I}_2$. In this case, we can essentially use notation and operations for matrices. For instance, we can perform singular value decomposition of $\mathcal{C}_{\mathscr{I}_1;\mathscr{I}_2} = \sum_{i=1}^{\infty} s_i(u_i \otimes v_i)$ where $s_i \in \mathbb{R}$ are ordered in nonincreasing manner, $\{u_i\}_{i=1}^{\infty} \subset \mathcal{F}^s$ and $\{v_i\}_{i=1}^{\infty} \subset \mathcal{F}^{d-s}$ are singular vectors. The rank of $\mathcal{C}_{\mathscr{I}_1;\mathscr{I}_2}$ is the smallest $r$ such that $s_i = 0$ for $i \geq r$. In this case, we will also define $\mathcal{U}_r = (u_1, u_2, \ldots, u_r)$, $\mathcal{V}_r = (v_1, v_2, \ldots, v_r)$ and $\mathcal{S}_r = \mathrm{diag}(s_1, s_2, \ldots, s_r)$, and denote the low rank approximation as $\mathcal{C}_{\mathscr{I}_1;\mathscr{I}_2} = \mathcal{U}_r \mathcal{S}_r \mathcal{V}_r^{\top}$. Finally, a 1st order tensor $reshape(\mathcal{C}_{X_{1:d}}, \{X_{1:d}\}, \emptyset)$, is simply a vector where we we will use vector notation.

## 4   Low Rank Kernel Embeddings Induced by Latent Variables

In the presence of latent variables, the kernel embedding $\mathcal{C}_{X_{1:d}}$ will be low rank. For example, the two observed variables $X_1$ and $X_2$ in the example in Figure 1 is conditional independent given the latent cluster indicator variable $Z$. That is the joint density factorizes as $p(X_1, X_2) = \sum_z p(z)p(X_1|z)p(X_2|z)$ (see Figure 2(a) for the graphical model). Throughout the paper, we assume that $z$ is discrete and takes $r$ possible values. Then the embedding $\mathcal{C}_{X_1 X_2}$ of $p(X_1, X_2)$ has a rank at most $r$. Let $z$ be represented as the standard basis in $\mathbb{R}^r$. Then

$$\mathcal{C}_{X_1 X_2} = \mathbb{E}_Z \left[ \left(\mathbb{E}_{X_1|Z}[\phi(X_1)]Z\right) \otimes \left(\mathbb{E}_{X_2|Z}[\phi(X_2)]Z\right)\right] = \mathcal{C}_{X_1|Z} \mathbb{E}_Z [Z \otimes Z] \left(\mathcal{C}_{X_2|Z}\right)^{\top} \quad (7)$$

where $\mathbb{E}_Z [Z \otimes Z]$ is an $r \times r$ matrix, and hence restricting the rank of $\mathcal{C}_{X_1 X_2}$ to be at most $r$.

In our second example, four observed variables are connected via two latent variables $Z_1$ and $Z_2$ each taking $r$ possible values. The conditional independence structure implies that the density of $p(X_1, X_2, X_3, X_4)$ factorizes as $\sum_{z_1, z_2} p(X_1|z_1)p(X_2|z_1)p(z_1, z_2)p(X_3|z_2)p(X_4|z_2)$ (see Figure 2(b) for the graphical model). Reshaping its kernel embedding $\mathcal{C}_{X_{1:4}}$, we obtain $\mathcal{C}_{X_{1:2};X_{3:4}} = reshape(\mathcal{C}_{X_{1:4}}, \{X_{1:2}\}, \{X_{3:4}\})$ which factorizes as

$$\mathbb{E}_{X_{1:2}|Z_1}[\phi(X_1) \otimes \phi(X_2)] \, \mathbb{E}_{Z_1 Z_2}[Z_1 \otimes Z_2] \, \left(\mathbb{E}_{X_{3:4}|Z_2}[\phi(X_3) \otimes \phi(X_4)]\right)^{\top} \quad (8)$$

where $\mathbb{E}_{Z_1 Z_2}[Z_1 \otimes Z_2]$ is an $r \times r$ matrix. Hence the intrinsic "rank" of the reshaped embedding is only $r$, although the original kernel embedding $\mathcal{C}_{X_{1:4}}$ is a 4th order tensor with infinite dimensions.

In general, for a latent variable model $p(X_1, \ldots, X_d)$ where the conditional independence structure is a tree $\mathcal{T}$, various reshapings of its kernel embedding $\mathcal{C}_{X_{1:d}}$ according to edges in the tree will be low rank. More specifically, each edge in the latent tree corresponds to a pair of latent variables $(Z_s, Z_t)$ (or an observed and a hidden variable $(X_s, Z_t)$) which induces a partition of the observed variables into two groups, $\mathscr{I}_1$ and $\mathscr{I}_2$. One can imagine splitting the latent tree into two subtrees by cutting the edge. One group of variables reside in the first subtree, and the other group in the second subtree. If we reshape the tensor according to this partitioning, then

**Theorem 1** *Assume that all observed variables are leaves in the latent tree structure, and all latent variables take $r$ possible values, then* $\mathrm{rank}(\mathcal{C}_{\mathscr{I}_1;\mathscr{I}_2}) \leq r$.

**Proof** Due to the conditional independence structure induced by the latent tree, $p(X_1, \ldots, X_d) = \sum_{z_s} \sum_{z_t} p(\mathscr{I}_1|z_s)p(z_s, z_t)p(\mathscr{I}_2|z_t)$. Then its embedding can be written as

$$\mathcal{C}_{\mathscr{I}_1;\mathscr{I}_2} = \mathcal{C}_{\mathscr{I}_1|Z_s} \mathbb{E}_{Z_s Z_t}[Z_s \otimes Z_t] \left(\mathcal{C}_{\mathscr{I}_2|Z_t}\right)^{\top}, \quad (9)$$

where $\mathcal{C}_{\mathscr{I}_1|Z_s}$ and $\mathcal{C}_{\mathscr{I}_2|Z_t}$ are the conditional embedding operators for $p(\mathscr{I}_1|z_s)$ and $p(\mathscr{I}_2|z_t)$ respectively. Since $\mathbb{E}_{Z_s Z_t}[Z_s \otimes Z_t]$ is a $r \times r$ matrix, $\mathrm{rank}(\mathcal{C}_{\mathscr{I}_1;\mathscr{I}_2}) \leq r$. ∎

Theorem 1 implies that, given a latent tree model, we obtain a collection of low rank reshapings $\{\mathcal{C}_{\mathscr{I}_1;\mathscr{I}_2}\}$ of the kernel embedding $\mathcal{C}_{X_{1:d}}$, each corresponding to an edge $(Z_s, Z_t)$ of the tree. We

will denote by $\mathcal{H}(\mathcal{T}, r)$ the class of kernel embeddings $\mathcal{C}_{X_{1:d}}$ whose various reshapings according to the latent tree $\mathcal{T}$ have rank at most $r$.[1] We will also use $\mathcal{C}_{X_{1:d}} \in \mathcal{H}(\mathcal{T}, r)$ to indicator such a relation.

In practice, the latent tree model assumption may be misspecified for a joint density $p(X_1, \ldots, X_d)$, and consequently the various reshapings of its kernel embedding $\mathcal{C}_{X_{1:d}}$ are only approximately low rank. In this case, we will instead impose a (potentially misspecified) latent structure $\mathcal{T}$ and a fixed rank $r$ on the data and obtain an approximate low rank decomposition of the kernel embedding. The goal is to obtain a low rank embedding $\widetilde{\mathcal{C}}_{X_{1:d}} \in \mathcal{H}(\mathcal{T}, r)$, while at the same time insure $\|\widetilde{\mathcal{C}}_{X_{1:d}} - \mathcal{C}_{X_{1:d}}\|_\bullet$ is small. In the following, we will present such a decomposition algorithm.

## 5  Low Rank Decomposition of Kernel Embeddings

For simplicity of exposition, we will focus on the case where the latent tree structure $\mathcal{T}$ has a caterpillar shape (Figure 2(c)). This decomposition can be viewed as a kernel generalization of the hierarchical tensor decomposition in [11, 12, 7]. The decomposition proceeds by reshaping the kernel embedding $\mathcal{C}_{X_{1:d}}$ according to the first edge $(Z_1, Z_2)$, resulting in $\mathcal{A}_1 := \mathcal{C}_{X_1; X_{2:d}}$. Then we perform a rank $r$ approximation for it, resulting in $\mathcal{A}_1 \approx \mathcal{U}_r \mathcal{S}_r \mathcal{V}_r^\top$. This leads to the first intermediate tensor $\mathcal{G}_1 = \mathcal{U}_r$, and we reshape $\mathcal{S}_r \mathcal{V}_r^\top$ and recursively decompose it. We note that Algorithm 1 contains only pseudo codes, and not implementable in practice since the kernel embedding to decompose can have infinite dimensions. We will design a practical kernel algorithm in the next section.

---

**Algorithm 1** Low Rank Decomposition of Kernel Embeddings

---

**In**: A kernel embedding $\mathcal{C}_{X_{1:d}}$, the caterpillar tree $\mathcal{T}$ and desired rank $r$

**Out**: A low rank embedding $\widetilde{\mathcal{C}}_{X_{1:d}} \in \mathcal{H}(\mathcal{T}, r)$ as intermediate tensors $\{\mathcal{G}_1, \ldots, \mathcal{G}_d\}$

1: $\mathcal{A}_1 = reshape(\mathcal{C}_{X_{1:d}}, \{X_1\}, \{X_{2:d}\})$ according to tree $\mathcal{T}$.
2: $\mathcal{A}_1 \approx \mathcal{U}_r \mathcal{S}_r \mathcal{V}_r^\top$, approximate $\mathcal{A}_1$ using its $r$ leading singular vectors.
3: $\mathcal{G}_1 = \mathcal{U}_r$, and $\mathcal{B}_1 = \mathcal{S}_r \mathcal{V}_r^\top$. $\mathcal{G}_1$ can be viewed as a model with two variables, $X_1$ and $Z_1$; and $\mathcal{B}_1$ as a new caterpillar tree model $\mathcal{T}_1$ with variable $X_1$ removed from $\mathcal{T}$.
4: **for** $j = 2, \ldots, d-1$ **do**
5:    $\mathcal{A}_j = reshape(\mathcal{B}_{j-1}, \{Z_{j-1}, X_j\}, \{X_{j+1:d}\})$ according to tree $\mathcal{T}_{j-1}$.
6:    $\mathcal{A}_j \approx \mathcal{U}_r \mathcal{S}_r \mathcal{V}_r^\top$, approximate $\mathcal{A}_j$ using its $r$ leading singular vectors.
7:    $\mathcal{G}_j = reshape(\mathcal{U}_r, \{Z_{j-1}\}, \{X_j\}, \{Z_j\})$, and $\mathcal{B}_j = \mathcal{S}_r \mathcal{V}_r^\top$. $\mathcal{G}_j$ can be viewed as a model with three variables, $X_j$, $Z_j$ and $Z_{j-1}$; and $\mathcal{B}_j$ as a new caterpillar tree model $\mathcal{T}_j$ with variable $Z_{j-1}$ and $X_j$ removed from $\mathcal{T}_{j-1}$.
8: **end for**
9: $\mathcal{G}_d = \mathcal{B}_{d-1}$

---

Once we finish the decomposition, we obtain the low rank representation of the kernel embedding as a set of intermediate tensors $\{\mathcal{G}_1, \ldots, \mathcal{G}_d\}$. In particular, we can think of $\mathcal{G}_1$ as a second order tensor with dimension $\infty \times r$, $\mathcal{G}_d$ as a second order tensor with dimension $r \times \infty$, and $\mathcal{G}_j$ for $2 \leqslant j \leqslant d-1$ as a third order tensor with dimension $r \times \infty \times r$. Then we can apply the low rank kernel embedding $\widetilde{\mathcal{C}}_{X_{1:d}}$ to a set of elements $\{f_i \in \mathcal{F}\}_{i=1}^d$ as follows $\widetilde{\mathcal{C}}_{X_{1:d}} \bullet_1 f_1 \bullet_2 \ldots \bullet_d f_d = (\mathcal{G}_1 \bullet_1 f_1)^\top (\mathcal{G}_2 \bullet_2 f_2) \ldots (\mathcal{G}_{d-1} \bullet_2 f_{d-1})(\mathcal{G}_d \bullet_2 f_d)$. Based on the above decomposition, one can obtain a low rank density estimate by $\widetilde{p}(X_1, \ldots, X_d) = \widetilde{\mathcal{C}}_{X_{1:d}} \bullet_1 \phi(X_1) \bullet_2 \ldots \bullet_d \phi(X_d)$. We can also compute the difference between $\widetilde{\mathcal{C}}_{X_{1:d}}$ and the operator $\mathcal{C}_{X_{1:d}}$ by using the generalized Frobenius norm $\|\widetilde{\mathcal{C}}_{X_{1:d}} - \mathcal{C}_{X_{1:d}}\|_\bullet$.

## 6  Kernel Algorithm

In practice, we are only provided with a finite number of samples $\{(x_1^i, \ldots, x_d^i)\}_{i=1}^n$ draw *i.i.d.* from $p(X_1, \ldots, X_d)$, and we want to obtain an empirical low rank decomposition of the kernel embedding. In this case, we will perform a low rank decomposition of the empirical kernel embedding $\bar{\mathcal{C}}_{X_{1:d}} = \frac{1}{n} \sum_{i=1}^n \left( \otimes_{j=1}^d \phi(x_j^i) \right)$. Although the empirical kernel embedding still has infinite dimensions, we will show that we can carry out the decomposition using just the kernel matrices. Let us denote the kernel matrix for each dimension of the data by $K_j$ where $j \in \{1, \ldots, d\}$. The $(i, i')$-th entry in $K_j$ can be computed as $K_j^{ii'} = k(x_j^i, x_j^{i'})$. Alternatively, one can think of implicitly forming

the feature matrix $\Phi_j = \left(\phi(x_j^1), \ldots, \phi(x_j^n)\right)$, and the corresponding kernel matrix is $K_j = \Phi_j^\top \Phi_j$. Furthermore, we denote the tensor feature matrix formed from dimension $j+1$ to $d$ of the data as $\Psi_j = \left(\otimes_{j'=j+1}^d \phi(x_{j'}^1), \ldots, \otimes_{j'=j+1}^d \phi(x_{j'}^n)\right)$. The corresponding kernel matrix $L_j = \Psi_j^\top \Psi_j$ with the $(i, i')$-th entry in $L_j$ defined as $L_j^{ii'} = \prod_{j'=j+1}^d k(x_{j'}^i, x_{j'}^{i'})$.

**Step 1-3 in Algorithm 1.** The key building block of the algorithm is a kernel singular value decomposition (Algorithm 2), which we will explain in more details using the example in step 2 of Algorithm 1. Using the implicitly defined feature matrix, $\mathcal{A}_1$ can be expressed as $\mathcal{A}_1 = \frac{1}{n}\Phi_1 \Psi_1^\top$. For the low rank approximation, $\mathcal{A}_1 \approx \mathcal{U}_r \mathcal{S}_r \mathcal{V}_r^\top$, using singular value decomposition, the leading $r$ singular vector $\mathcal{U}_r = (u_1, \ldots, u_r)$ will lie in the span of $\Phi_1$, *i.e.*, $\mathcal{U}_r = \Phi_1(\boldsymbol{\beta}_1, \ldots, \boldsymbol{\beta}_r)$ where $\boldsymbol{\beta} \in \mathbb{R}^n$. Then we can transform the singular value decomposition problem for an infinite dimensional matrix to a generalized eigenvalue problem involving kernel matrices, $\mathcal{A}_1 \mathcal{A}_1^\top u = \lambda\, u \Leftrightarrow \frac{1}{n^2}\Phi_1 \Psi_1^\top \Psi_1 \Phi_1^\top \Phi_1 \boldsymbol{\beta} = \lambda \Phi_1 \boldsymbol{\beta} \Leftrightarrow \frac{1}{n^2} K_1 L_1 K_1 \boldsymbol{\beta} = \lambda K_1 \boldsymbol{\beta}$. Let the Cholesky decomposition of $K_1$ be $R^\top R$, then the generalized eigenvalue decomposition problem can be solved by redefining $\widetilde{\boldsymbol{\beta}} = R\boldsymbol{\beta}$, and solving an ordinary eigenvalue problem

$$\frac{1}{n^2} R L_1 R^\top \widetilde{\boldsymbol{\beta}} = \lambda \widetilde{\boldsymbol{\beta}}, \text{ and obtain } \boldsymbol{\beta} = R^\dagger \widetilde{\boldsymbol{\beta}}. \tag{10}$$

The resulting singular vectors satisfy $u_l^\top u_{l'} = \boldsymbol{\beta}_l^\top \Phi_1^\top \Phi_1 \boldsymbol{\beta}_{l'} = \boldsymbol{\beta}_l^\top K \boldsymbol{\beta}_{l'} = \widetilde{\boldsymbol{\beta}}_l^\top \widetilde{\boldsymbol{\beta}}_{l'} = \delta_{ll'}$. Then we can obtain $\mathcal{B}_1 := \mathcal{S}_r \mathcal{V}_r^\top = \mathcal{U}_r^\top \mathcal{A}_1$ by projecting the column of $\mathcal{A}_1$ using the singular vectors $\mathcal{U}_r$,

$$\mathcal{B}_1 = \frac{1}{n}(\boldsymbol{\beta}_1, \ldots, \boldsymbol{\beta}_r)^\top \Phi_1^\top \Phi_1 \Psi_1^\top = \frac{1}{n}(\boldsymbol{\beta}_1, \ldots, \boldsymbol{\beta}_r)^\top K_1 \Psi_1^\top =: (\boldsymbol{\gamma}^1, \ldots, \boldsymbol{\gamma}^n)\Psi_1^\top \tag{11}$$

where $\boldsymbol{\gamma} \in \mathbb{R}^r$ can be treated as the reduced $r$-dimensional feature representation for each feature mapped data point $\phi(x_1^i)$. Then we have the first intermediate tensor $\mathcal{G}_1 = \mathcal{U}_r = \Phi_1(\boldsymbol{\beta}_1, \ldots, \boldsymbol{\beta}_r) =: \Phi_1(\boldsymbol{\theta}^1, \ldots, \boldsymbol{\theta}^n)^\top$, where $\boldsymbol{\theta} \in \mathbb{R}^r$. Then the kernel singular value decomposition can be carried out recursively on the reshaped tensor $\mathcal{B}_1$.

**Step 5-7 in Algorithm 1.** When $j = 2$, we first reshape $\mathcal{B}_1 = \mathcal{S}_r \mathcal{V}_r^\top$ to obtain $\mathcal{A}_2 = \frac{1}{n}\widetilde{\Phi}_2 \Psi_2^\top$, where $\widetilde{\Phi}_2 = (\boldsymbol{\gamma}^1 \otimes \phi(x_2^1), \ldots, \boldsymbol{\gamma}^n \otimes \phi(x_2^n))$. Then we can carry out similar singular value decomposition as before, and obtain $\mathcal{U}_r = \widetilde{\Phi}_2(\boldsymbol{\beta}_1, \ldots, \boldsymbol{\beta}_r) =: \widetilde{\Phi}_2(\boldsymbol{\theta}^1, \ldots, \boldsymbol{\theta}^n)^\top$. Then we have the second operator $\mathcal{G}_2 = \sum_{i=1}^n \boldsymbol{\gamma}^i \otimes \phi(x_2^i) \otimes \boldsymbol{\theta}^i$. Last, we define $\mathcal{B}_2 := \mathcal{S}_r \mathcal{V}_r^\top = \mathcal{U}_r^\top \mathcal{A}_2$ as

$$\mathcal{B}_2 = \frac{1}{n}(\boldsymbol{\beta}_1, \ldots, \boldsymbol{\beta}_r)^\top \widetilde{\Phi}_2^\top \widetilde{\Phi}_2 \Psi_2^\top = \frac{1}{n}(\boldsymbol{\beta}_1, \ldots, \boldsymbol{\beta}_r)^\top (\Gamma \circ K_2)\Psi_2^\top =: \frac{1}{n}(\boldsymbol{\gamma}^1, \ldots, \boldsymbol{\gamma}^n)\Psi_2^\top, \tag{12}$$

and carry out the recursive decomposition further.

The result of the algorithm is an empirical low rank kernel embedding, $\widehat{\mathcal{C}}_{X_{1:d}}$, represented as a collection of intermediate tensors $\{\mathcal{G}_1, \ldots, \mathcal{G}_d\}$. The overall algorithm is summarized in Algorithm 3. More details about the derivation can be found in Appendix A.

The application of the set of intermediate tensor $\{\mathcal{G}_1, \ldots, \mathcal{G}_d\}$ to a set of elements $\{f_i \in \mathcal{F}\}$ can be expressed as kernel operations. For instance, we can obtain a density estimate by $\widehat{p}(x_1, \ldots, x_d) = \widehat{\mathcal{C}}_{X_{1:d}} \bullet_1 \phi(x_1) \bullet_2 \ldots \bullet_d \phi(x_d) = \sum_{z_1, \ldots, z_d} g_1(x_1, z_1) g_2(z_1, x_2, z_2) \ldots g_d(z_{d-1}, x_d)$ where (see Appendix A for more details)

$$g_1(x_1, z_1) = \mathcal{G}_1 \bullet_1 \phi(x_1) \bullet_2 z_1 = \sum_{i=1}^n (z_1^\top \boldsymbol{\theta}^i) k(x_1^i, x_1) \tag{13}$$

$$g_j(z_{j-1}, x_j, z_j) = \mathcal{G}_j \bullet_1 z_{j-1} \bullet_2 \phi(x_j) \bullet_3 z_j = \sum_{i=1}^n (z_{j-1}^\top \boldsymbol{\gamma}^i) k(x_j^i, x_j)(z_j^\top \boldsymbol{\theta}^i) \tag{14}$$

$$g_d(z_{d-1}, x_d) = \mathcal{G}_d \bullet_1 z_{d-1} \bullet x_d = \sum_{i=1}^n (z_{d-1}^\top \boldsymbol{\gamma}^i) k(x_d^i, x_d) \tag{15}$$

In the above formulas, each term is a weighted combination of kernel functions, and the weighting is determined by the kernel singular value decomposition and the values of the latent variable $\{z_j\}$.

## 7  Performance Guarantees

As we mentioned in the introduction, the imposed latent structure used in the low rank decomposition of kernel embeddings may be misspecified, and the decomposition of empirical embeddings may suffer from sampling error. In this section, we provide finite guarantee for Algorithm 3 even when the latent structures are misspecified. More specifically, we will bound, in terms of the gen-

---
**Algorithm 2** KernelSVD$(K, L, r)$

---
**Out**: A collection of vectors $(\boldsymbol{\theta}^1, \ldots, \boldsymbol{\theta}^n)$
 1: Perform Cholesky decomposition $K = R^\top R$
 2: Solve eigen decomposition $\frac{1}{n^2} R L R^\top \widetilde{\boldsymbol{\beta}} = \lambda \widetilde{\boldsymbol{\beta}}$, and keep the leading $r$ eigen vectors $(\widetilde{\boldsymbol{\beta}}_1, \ldots, \widetilde{\boldsymbol{\beta}}_r)$
 3: Compute $\boldsymbol{\beta}_1 = R^\dagger \widetilde{\boldsymbol{\beta}}_1, \ldots, \boldsymbol{\beta}_r = R^\dagger \widetilde{\boldsymbol{\beta}}_r$, and reorgnaize $(\boldsymbol{\theta}^1, \ldots, \boldsymbol{\theta}^n)^\top = (\boldsymbol{\beta}_1, \ldots, \boldsymbol{\beta}_r)$

---

---
**Algorithm 3** Kernel Low Rank Decomposition of Empirical Embedding $\bar{\mathcal{C}}_{X_{1:d}}$

---
**In**: A sample $\left\{(x_1^i, \ldots, x_d^i)\right\}_{i=1}^n$, desired rank $r$, a query point $(x_1, \ldots, x_d)$
**Out**: A low rank embedding $\widehat{\mathcal{C}}_{X_{1:d}} \in \mathcal{H}(\mathcal{T}, r)$ as intermediate operators $\{\mathcal{G}_1, \ldots, \mathcal{G}_d\}$
 1: $L_d = \mathbf{1}\mathbf{1}^\top$
 2: **for** $j = d, d-1, \ldots, 1$ **do**
 3:    Compute matrix $K_j$ with $K_j^{ii'} = k(x_j^i, x_j^{i'})$; furthermore, if $j < d$, then $L_j = L_{j+1} \circ K_{j+1}$
 4: **end for**
 5: $(\boldsymbol{\theta}^1, \ldots, \boldsymbol{\theta}^n) = \text{KernelSVD}(K_1, L_1, r)$
 6: $\mathcal{G}_1 = \Phi_1(\boldsymbol{\theta}^1, \ldots, \boldsymbol{\theta}^n)^\top$, and compute $(\boldsymbol{\gamma}^1, \ldots, \boldsymbol{\gamma}^n) = (\boldsymbol{\theta}^1, \ldots, \boldsymbol{\theta}^n) K_1$
 7: **for** $j = 2, \ldots, d-1$ **do**
 8:    $\Gamma = (\boldsymbol{\gamma}^1, \ldots, \boldsymbol{\gamma}^n)^\top (\boldsymbol{\gamma}^1, \ldots, \boldsymbol{\gamma}^n)$, and compute $(\boldsymbol{\theta}^1, \ldots, \boldsymbol{\theta}^n) = \text{KernelSVD}(K_i \circ \Gamma, L_i, r)$
 9:    $\mathcal{G}_j = \sum_{i=1}^n \boldsymbol{\gamma}^i \otimes \phi(x_j^i) \otimes \boldsymbol{\theta}^i$, and compute $(\boldsymbol{\gamma}^1, \ldots, \boldsymbol{\gamma}^n) = (\boldsymbol{\theta}^1, \ldots, \boldsymbol{\theta}^n) K_i$
10: **end for**
11: $\mathcal{G}_d = (\boldsymbol{\gamma}^1, \ldots, \boldsymbol{\gamma}^n) \Phi_d^\top$

---

eralized Frobenius norm $\|\mathcal{C}_{X_{1:d}} - \widehat{\mathcal{C}}_{X_{1:d}}\|_\bullet$, the difference between the true kernel embeddings and the low rank kernel embeddings estimated from a set of $n$ *i.i.d.* samples $\left\{(x_1^i, \ldots, x_d^i)\right\}_{i=1}^n$. First we observed that the difference can be decomposed into two terms

$$\|\mathcal{C}_{X_{1:d}} - \widehat{\mathcal{C}}_{X_{1:d}}\|_\bullet \leqslant \underbrace{\|\mathcal{C}_{X_{1:d}} - \widetilde{\mathcal{C}}_{X_{1:d}}\|_\bullet}_{E_1: \text{ model error}} + \underbrace{\|\widetilde{\mathcal{C}}_{X_{1:d}} - \widehat{\mathcal{C}}_{X_{1:d}}\|_\bullet}_{E_2: \text{ estimation error}} \tag{16}$$

where the first term is due to the fact that the latent structures may be misspecified, while the second term is due to estimation from finite number of data points. We will bound these two sources of error separately (the proof is deferred to Appendix B)

**Theorem 2** *Suppose each reshaping $\mathcal{C}_{\mathscr{I}_1;\mathscr{I}_2}$ of $\mathcal{C}_{X_{1:d}}$ according to an edge in the latent tree structure has a rank $r$ approximation $\mathcal{U}_r \mathcal{S}_r \mathcal{V}_r^\top$ with error $\left\|\mathcal{C}_{\mathscr{I}_1;\mathscr{I}_2} - \mathcal{U}_r \mathcal{S}_r \mathcal{V}_r^\top\right\|_\bullet \leqslant \epsilon$. Then the low rank decomposition $\widetilde{\mathcal{C}}_{X_{1:d}}$ from Algorithm 1 satisfies $\|\mathcal{C}_{X_{1:d}} - \widetilde{\mathcal{C}}_{X_{1:d}}\|_\bullet \leqslant \sqrt{d-1}\,\epsilon$.*

Although previous work [5, 6] have also used hierarchical decomposition for kernel embeddings, their decompositions make the strong assumption that the latent tree models are correctly specified. When the models are misspecified, these algorithms have no guarantees whatsoever, and may fail drastically as we show in later experiments. In contrast, the decomposition we proposed here are robust in the sense that even when the latent tree structure is misspecified, we can still provide the approximation guarantee for the algorithm. Furthermore, when the latent tree structures are correctly specified and the rank $r$ is also correct, then $\mathcal{C}_{\mathscr{I}_1;\mathscr{I}_2}$ has rank $r$ and hence $\epsilon = 0$ and our decomposition algorithm does not incur any modeling error.

Next, we provide bound for the the estimation error. The estimation error arises from decomposing the empirical estimate $\bar{\mathcal{C}}_{X_{1:d}}$ of the kernel embedding, and the error can accumulate as we combine intermediate tensors $\{\mathcal{G}_1, \ldots, \mathcal{G}_d\}$ to form the final low rank kernel embedding. More specifically, we have the following bound (the proof is deferred to Appendix C)

**Theorem 3** *Suppose the $r$-th singular value of each reshaping $\mathcal{C}_{\mathscr{I}_1;\mathscr{I}_2}$ of $\mathcal{C}_{X_{1:d}}$ according to an edge in the latent tree structure is lower bounded by $\lambda$, then with probability at least $1 - \delta$, $\|\widetilde{\mathcal{C}}_{X_{1:d}} - \widehat{\mathcal{C}}_{X_{1:d}}\|_\bullet \leqslant \frac{(1+\lambda)^{d-2}}{\lambda^{d-2}} \|\mathcal{C}_{X_{1:d}} - \bar{\mathcal{C}}_{X_{1:d}}\|_\bullet \leqslant \frac{(1+\lambda)^{d-2}c}{\lambda^{d-2}\sqrt{n}}$, with some constant $c$ associated with the kernel and the probability $\delta$.*

From the above theorem, we can see that the smaller the $r$-th singular value, the more difficult it is to estimate the low rank kernel embedding. Although in the bound the error grows exponential in $1/\lambda^{d-2}$, in our experiments, we did not observe such exponential degradation of performance even in relatively high dimensional datasets.

## 8    Experiments

Besides the synthetic dataset we showed in Figure 1 where low rank kernel embedding can lead to significant improvement in term of estimating the density, we also experimented with real world datasets from UCI data repository. We take 11 datasets with varying dimensions and number of data points, and the attributes of the datasets are continuous-valued. We whiten the data and compare low rank kernel embeddings (Low Rank) obtained from Algorithm 3 to 3 other alternatives for continuous density estimation, namely, mixture of Gaussian with full covariance matrix, ordinary kernel density estimator (KDE) and the kernel spectral algorithm for latent trees (Spectral) [6]. We use Gaussian kernel $k(x, x') = \frac{1}{\sqrt{2\pi}s}\exp(-\|x - x'\|^2/(2s^2))$ for KDE, Spectral and our method (Low rank). We split each dataset into 10 subsets, and use nested cross-validation based on held-out likelihood to choose hyperparameters: the kernel parameter $s$ for KDE, Spectral and Low rank ($\{2^{-3}, 2^{-2}, 2^{-1}, 1, 2, 4, 8\}$ times the median pairwise distance), the rank parameter $r$ for Spectral and Low rank (range from 2 to 30), and the number of components in the Gaussian mixture (range from 2 to $\frac{\texttt{\# Sample}}{30}$). For both Spectral and Low rank, we use a caterpillar tree in Figure 2(c) as the structure for the latent variable model.

From Table 1, we can see that low rank kernel embeddings provide the best or comparable held-out negative log-likelihood across the datasets we experimented with. In some datasets, low rank kernel embeddings can lead to drastic improvement over the alternatives. For instance, in dataset "sonar" and "yeast", the improvement is dramatic. The Spectral approach performs even worse sometimes. This makes sense, since the caterpillar tree supplied to the algorithm may be far away from the reality and Spectral is not robust to model misspecification. Meanwhile, the Spectral algorithm also caused numerical problem in practical. In contrast, our method Low Rank uses the same latent structure, but achieved much more robust results.

Table 1: Negative log-likelihood on held-out data (the lower the better).

| Data Set | # Sample | Dim. | Gaussian mixture | KDE | Spectral | Low rank |
|---|---|---|---|---|---|---|
| australian | 690 | 14 | 17.97±0.26 | 18.32±0.64 | 33.50 ±2.17 | **15.88±0.11** |
| bupa | 345 | 6 | 8.17±0.30 | 8.36±0.17 | 25.01±0.66 | **7.57±0.14** |
| german | 1000 | 24 | 31.14 ± 0.41 | 30.57 ± 0.15 | 28.40 ± 11.64 | **22.89 ± 0.26** |
| heart | 270 | 13 | 17.72 ±0.23 | 18.23 ±0.18 | 21.50 ± 2.39 | **16.95 ± 0.13** |
| ionosphere | 351 | 34 | 47.60 ±1.77 | 43.53 ± 1.25 | 54.91±1.35 | **35.84 ± 1.00** |
| pima | 768 | 8 | 11.78 ± 0.04 | 10.38 ± 0.19 | 31.42 ± 2.40 | **10.07 ± 0.11** |
| parkinsons | 195 | 22 | 30.13± 0.24 | 30.65 ± 0.66 | 33.20 ± 0.70 | **28.19 ± 0.37** |
| sonar | 208 | 60 | 107.06 ± 1.36 | 96.17 ± 0.27 | 89.26 ± 2.75 | **57.96 ± 2.67** |
| wpbc | 198 | 33 | 50.75 ± 1.11 | 49.48 ± 0.64 | 48.66 ± 2.56 | **40.78 ± 0.86** |
| wine | 178 | 13 | 19.59 ± 0.14 | 19.56 ± 0.56 | 19.25 ± 0.58 | **18.67 ± 0.17** |
| yeast | 208 | 79 | 146.11 ± 5.36 | 137.15 ± 1.80 | 76.58 ± 2.24 | **72.67 ±4.05** |

## 9    Discussion and Conclusion

In this paper, we presented a robust kernel embedding algorithm which can make use of the low rank structure of the data, and provided both theoretical and empirical support for it. However, there are still a number of issues which deserve further research. First, the algorithm requires a sequence of kernel singular decompositions which can be computationally intensive for high dimensional and large datasets. Developing efficient algorithms yet with theoretical guarantees will be interesting future research. Second, the statistical analysis could be sharpened. For the moment, the analysis does not seem to suggest that the obtained estimator by our algorithm is better than ordinary KDE. Third, it will be interesting empirical work to explore other applications for low rank kernel embeddings, such as kernel two-sample tests, kernel independence tests and kernel belief propagation.

## Footnotes

[1]One can readily generalize this notation to decompositions where different reshapings have different ranks.

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
