[Supplementary Material · appendix.pdf]

# A  Kernel Algorithm

In practice, we are only provided with a finite number of samples $\left\{(x_1^i, \ldots, x_d^i)\right\}_{i=1}^n$ draw *i.i.d.* from $p(X_1, \ldots, X_d)$, and we want to obtain an empirical low rank decomposition of the kernel embedding. In this case, we will perform a low rank decomposition of the empirical kernel embedding $\widehat{\mathcal{C}}_{X_{1:d}} = \frac{1}{n} \sum_{i=1}^n \left(\otimes_{j=1}^d \phi(x_j^i)\right)$. Although the empirical kernel embedding still has infinite dimensions, we will show that we can carry out the decomposition using just the kernel matrices.

Let us denote the kernel matrix for each dimension of the data by $K_j$ where $j \in \{1, \ldots, d\}$. The $(i, i')$-th entry in $K_j$ can be computed as $K_j^{ii'} = k(x_j^i, x_j^{i'})$. Alternatively, one can think of implicitly forming the feature matrix $\Phi_j = \left(\phi(x_j^1), \ldots, \phi(x_j^n)\right)$, and the corresponding kernel matrix is $K_j = \Phi_j^\top \Phi_j$. Furthermore, we denote the tensor feature matrix formed from dimension $j+1$ to $d$ of the data as $\Psi_j = \left(\otimes_{j'=j+1}^d \phi(x_{j'}^1), \ldots, \otimes_{j'=j+1}^d \phi(x_{j'}^n)\right)$. The corresponding kernel matrix $L_j = \Psi_j^\top \Psi_j$ with the $(i, i')$-th entry in $L_j$ defined as $L_j^{ii'} = \prod_{j'=j+1}^d k(x_{j'}^i, x_{j'}^{i'})$. The overall kernel algorithm for low rank decomposition of empirical embeddings for $\widehat{\mathcal{C}}_{X_{1:d}}$ in terms of kernels is summarized in algorithm 3.

**For step 1-3 in Algorithm 1:** Using the implicitly defined feature matrix, $\mathcal{A}_1$ can be expressed as $\mathcal{A}_1 = \frac{1}{n} \Phi_1 \Psi_1^\top$. For the low rank approximation, $\mathcal{A}_1 \approx \mathcal{U}_r \mathcal{S}_r \mathcal{V}_r^\top$, using singular value decomposition, the leading $r$ singular vector $\mathcal{U}_r = (u_1, \ldots, u_r)$ will lie in the span of $\Phi_1$, *i.e.*, $\mathcal{U}_r = \Phi_1(\boldsymbol{\beta}_1, \ldots, \boldsymbol{\beta}_r)$ where $\boldsymbol{\beta} \in \mathbb{R}^n$. Then we can transform the singular value decomposition problem for an infinite dimensional matrix to a generalized eigenvalue problem involving kernel matrices, $\mathcal{A}_1 \mathcal{A}_1^\top u = \lambda\, u \Leftrightarrow \frac{1}{n^2} \Phi_1 \Psi_1^\top \Psi_1 \Phi_1^\top \Phi_1 \boldsymbol{\beta} = \lambda\, \Phi_1 \boldsymbol{\beta} \Leftrightarrow \frac{1}{n^2} K_1 L_1 K_1 \boldsymbol{\beta} = \lambda\, K_1 \boldsymbol{\beta}$. Let the Cholesky decomposition of $K_1$ be $K_1 = R^\top R$, then the generalized eigenvalue decomposition problem can be solved by redefining $\widetilde{\boldsymbol{\beta}} = R\boldsymbol{\beta}$, and solving an ordinary eigenvalue problem

$$\frac{1}{n^2} R L_1 R^\top \widetilde{\boldsymbol{\beta}} = \lambda\, \widetilde{\boldsymbol{\beta}}, \text{ and obtain } \boldsymbol{\beta} = R^\dagger \widetilde{\boldsymbol{\beta}}. \tag{17}$$

This procedure (Algorithm 2) satisfies $u_l^\top u_{l'} = \boldsymbol{\beta}_l^\top \Phi_1^\top \Phi_1 \boldsymbol{\beta}_{l'} = \boldsymbol{\beta}_l^\top K \boldsymbol{\beta}_{l'} = \boldsymbol{\beta}_l^\top R^\top R \boldsymbol{\beta}_{l'} = \widetilde{\boldsymbol{\beta}}_l^\top \widetilde{\boldsymbol{\beta}}_{l'} = \delta_{ll'}$. Then we can obtain $\mathcal{S}_r \mathcal{V}_r^\top$ by projecting the column of $\mathcal{A}_1$ using the singular vectors $\mathcal{U}_r$,

$$\mathcal{S}_r \mathcal{V}_r^\top = \mathcal{U}_r^\top \mathcal{A}_1 = \frac{1}{n} (\boldsymbol{\beta}_1, \ldots, \boldsymbol{\beta}_r)^\top \Phi_1^\top \Phi_1 \Psi_1^\top = \frac{1}{n} (\boldsymbol{\beta}_1, \ldots, \boldsymbol{\beta}_r)^\top K_1 \Psi_1^\top =: \frac{1}{n} (\boldsymbol{\gamma}^1, \ldots, \boldsymbol{\gamma}^n) \Psi_1^\top \tag{18}$$

where $\boldsymbol{\gamma} \in \mathbb{R}^r$ can be treated as the reduced $r$-dimensional feature representation for each feature mapped data point $\phi(x_1^i)$. Then we have the first intermediate operator $\mathcal{G}_1 = \mathcal{U}_r = \Phi_1(\boldsymbol{\beta}_1, \ldots, \boldsymbol{\beta}_r) =: \Phi_1(\boldsymbol{\theta}^1, \ldots, \boldsymbol{\theta}^n)^\top$, where $\boldsymbol{\theta} \in \mathbb{R}^n$. Furthermore we have

$$\mathcal{G}_1 \bullet_1 \phi(x_1) \bullet_2 z_1 = \phi(x_1)^\top \Phi_1(\boldsymbol{\theta}^1, \ldots, \boldsymbol{\theta}^n)^\top z_1 = \sum_{i=1}^n (z_1^\top \boldsymbol{\theta}^i) k(x_1^i, x_1).$$

which is a weighted combination of kernel functions, and the weighting is determined by the kernel singular value decomposition and the value of the latent variable $z_1$.

**For the first iteration in step 4-9:** After we obtain $\mathcal{G}_1$, we reshape $\mathcal{S}_r \mathcal{V}_r^\top$ to obtain the updated $\mathcal{B}_2 = \frac{1}{n} \widetilde{\Phi}_2 \Psi_2^\top$, where $\widetilde{\Phi}_2 = (\boldsymbol{\gamma}^1 \otimes \phi(x_2^1), \ldots, \boldsymbol{\gamma}^n \otimes \phi(x_2^n))$. Then we can carry out similar singular value decomposition as before, and obtain $\mathcal{U}_r = \widetilde{\Phi}_2(\boldsymbol{\beta}_1, \ldots, \boldsymbol{\beta}_r) =: \widetilde{\Phi}_2(\boldsymbol{\theta}^1, \ldots, \boldsymbol{\theta}^n)^\top$. Reshaping this operator, we have the second operator $\mathcal{G}_2 = reshape(\mathcal{U}_r, \{Z_1\}, \{X_2\}, \{Z_2\})$, and furthermore, when we have

$$\mathcal{G}_2 \bullet_1 z_1 \bullet_2 \phi(x_2) \bullet_3 z_2 = z_1^\top (\boldsymbol{\gamma}^1, \ldots, \boldsymbol{\gamma}^n) \operatorname{diag}(\Phi_2^\top \phi(x_2))(\boldsymbol{\theta}^1, \ldots, \boldsymbol{\theta}^n)^\top z_2 \tag{19}$$

$$= \sum_{i=1}^n (z_2^\top \boldsymbol{\gamma}^i) k(x_2^i, x_2)(z_3^\top \boldsymbol{\theta}^i). \tag{20}$$

And we can re-define the following as $\mathcal{B}_2$ and move on to the next iteration

$$\mathcal{S}_r\mathcal{V}_r^\top = \mathcal{U}_r^\top\mathcal{B}_2 = \frac{1}{n}(\boldsymbol{\beta}_1,\ldots,\boldsymbol{\beta}_r)^\top\widetilde{\Phi}_2^\top\widetilde{\Phi}_2\Psi_2^\top = \frac{1}{n}(\boldsymbol{\beta}_1,\ldots,\boldsymbol{\beta}_r)^\top(\Gamma\circ K_2)\Psi_2^\top =: \frac{1}{n}(\boldsymbol{\gamma}^1,\ldots,\boldsymbol{\gamma}^n)\Psi_2^\top, \tag{21}$$

## B  Bounding Model Error

**Theorem 1** *Suppose each reshaping $\mathcal{C}_{\mathscr{I}_1;\mathscr{I}_2}$ of $\mathcal{C}_{X_{1:d}}$ according to an edge in the latent tree structure a rank $r$ approximation $\mathcal{U}_r\mathcal{S}_r\mathcal{V}_r^\top$ with error $\left\|\mathcal{C}_{\mathscr{I}_1;\mathscr{I}_2} - \mathcal{U}_r\mathcal{S}_r\mathcal{V}_r^\top\right\|_\bullet \leqslant \epsilon$. Then the low rank decomposition $\widetilde{\mathcal{C}}_{X_{1:d}}$ from Algorithm 1 satisfies*

$$\left\|\mathcal{C}_{X_{1:d}} - \widetilde{\mathcal{C}}_{X_{1:d}}\right\|_\bullet \leqslant \sqrt{d-1}\,\epsilon \tag{22}$$

**Proof**  In the decomposition, $\mathcal{A}_1 = \mathcal{C}_{X_1;X_{2:d}}$ is approximated by $\mathcal{U}_r\mathcal{S}_r\mathcal{V}_r^\top$ where $\mathcal{B}_1 = \mathcal{S}_r\mathcal{V}_r^\top$ are reshaped and further approximated as $\mathcal{T}$. Suppose that $\mathcal{T}_1 = reshape(\mathcal{T},\{Z_1\},\{X_{2:d}\})$. Then the model error is bounded as

$$\left\|\mathcal{C}_{X_1;X_{2:d}} - \mathcal{U}_r^1\mathcal{T}_1\right\|_\bullet^2 \tag{23}$$

$$= \left\|\mathcal{A}_1 - \mathcal{U}_r^1(\mathcal{B}_1 - \mathcal{B}_1 + \mathcal{T}_1)\right\|_\bullet^2 \tag{24}$$

$$= \left\|\mathcal{A}_1 - \mathcal{U}_r^1\mathcal{B}_1\right\|_\bullet^2 + 2\left\langle\mathcal{A}_1 - \mathcal{U}_r^1\mathcal{B}_1, \mathcal{U}_r^1(\mathcal{B}_1 - \mathcal{T}_1)\right\rangle_\bullet^2 + \left\|\mathcal{U}_r^1(\mathcal{B}_1 - \mathcal{T}_1)\right\|_\bullet^2 \tag{25}$$

$$\leqslant \epsilon + 0 + \left\|\mathcal{B}_1 - \mathcal{T}_1\right\|_\bullet^2 \qquad (\mathcal{A}_1 - \mathcal{U}_r^1\mathcal{B}_1 \text{ is perpendicular to } \mathcal{U}_r) \tag{26}$$

$$= \epsilon^2 + \left\|\mathcal{A}_2 - \mathcal{U}_r^2\mathcal{T}_2\right\|_\bullet^2 \qquad (\text{reshaped tensors have the same generalized Frobenius norm}) \tag{27}$$

$$= \epsilon^2 + \left\|\mathcal{A}_2 - \mathcal{U}_r^2\mathcal{B}_2\right\|_\bullet^2 + \left\|\mathcal{B}_2 - \mathcal{T}_2\right\|_\bullet^2 \qquad (\mathcal{A}_2 - \mathcal{U}_r^2\mathcal{B}_2 \text{ is perpendicular to } \mathcal{U}_r^2) \tag{28}$$

$$= \epsilon^2 + \left\|(\mathcal{U}_r^{1\top}\otimes\mathcal{I})\mathcal{C}_{X_{1:2};X_{3:d}} - \mathcal{U}_r^2\mathcal{B}_2\right\|_\bullet + \left\|\mathcal{B}_2 - \mathcal{T}_2\right\|_\bullet^2 \qquad (\text{rewrite } \mathcal{A}_2) \tag{29}$$

$$\leqslant \epsilon^2 + \epsilon^2 + \left\|\mathcal{B}_2 - \mathcal{T}_2\right\|_\bullet^2 \qquad (\text{projection by } \mathcal{U}_r^{1\top}\otimes\mathcal{I} \text{ can only decrease singular value}) \tag{30}$$

$$\leqslant 2\epsilon^2 + \left\|((\mathcal{U}_r^{2\top}(\mathcal{U}_r^{1\top}\otimes\mathcal{I})\otimes\mathcal{I})\mathcal{C}_{X_{1:3};X_{4:d}} - \mathcal{U}_r^3\mathcal{B}_3\right\|_\bullet^2 + \left\|\mathcal{B}_3 - \mathcal{T}_3\right\|_\bullet^2 \qquad (\text{rewrite } \mathcal{A}_3) \tag{31}$$

$$\leqslant 3\epsilon^2 + \left\|\mathcal{B}_3 - \mathcal{T}_3\right\|_\bullet^2 \qquad (\text{projection can only decrease singular value}) \tag{32}$$

$$\leqslant (d-1)\epsilon^2 \qquad (\text{by induction on } i) \tag{33}$$

Since $\widetilde{\mathcal{C}}_{X_1;X_{2:d}} = \mathcal{U}_r^1\mathcal{T}_1$, we have that

$$\left\|\mathcal{C}_{X_1;X_{2:d}} - \widetilde{\mathcal{C}}_{X_1;X_{2:d}}\right\|_\bullet \leqslant \sqrt{d-1}\,\epsilon \quad \text{or} \quad \left\|\mathcal{C}_{X_{1:d}} - \widetilde{\mathcal{C}}_{X_{1:d}}\right\|_\bullet \leqslant \sqrt{d-1}\,\epsilon \tag{34}$$

∎

## C  Bounding Estimation Error

**Theorem 2** *Suppose the $r$-th singular value of each reshaping $\mathcal{C}_{\mathscr{I}_1;\mathscr{I}_2}$ of $\mathcal{C}_{X_{1:d}}$ according to an edge in the latent tree structure is lower bounded by $\lambda$, then with probability at least $1 - \delta$,*

$$\left\|\widetilde{\mathcal{C}}_{X_{1:d}} - \widehat{\mathcal{C}}_{X_{1:d}}\right\|_\bullet \leq \frac{(1+\lambda)^{d-2}}{\lambda^{d-2}}\left\|\mathcal{C}_{X_{1:d}} - \bar{\mathcal{C}}_{X_{1:d}}\right\|_\bullet \leqslant \frac{(1+\lambda)^{d-2}c}{\lambda^{d-2}\sqrt{n}} \tag{35}$$

*with some constant $c$ associated with the kernel and the probability $\delta$.*

**Proof** $\widehat{\mathcal{C}}_{X_{1:d}}$ is the empirical low rank embedding, a finite sample estimate for $\widetilde{\mathcal{C}}_{X_{1:d}}$. There difference can be bounded as

$$\left\|\widetilde{\mathcal{C}}_{X_{1:d}} - \widehat{\mathcal{C}}_{X_{1:d}}\right\|_{\bullet} \tag{36}$$

$$= \left\|\mathcal{U}_r^1 \mathcal{T}_1 - \widehat{\mathcal{U}}_r^1 \widehat{\mathcal{T}}_1\right\|_{\bullet} \tag{37}$$

$$\leqslant \left\|(\mathcal{U}_r^1 - \widehat{\mathcal{U}}_r^1)\mathcal{T}_1\right\|_{\bullet} + \left\|\widehat{\mathcal{U}}_r^1(\mathcal{T}_1 - \widehat{\mathcal{T}}_1)\right\|_{\bullet} \qquad \text{(triangular inequality)} \tag{38}$$

$$\leqslant \left\|\mathcal{U}_r^1 - \widehat{\mathcal{U}}_r^1\right\|_{\bullet} + \left\|\mathcal{T}_1 - \widehat{\mathcal{T}}_1\right\|_{\bullet} \qquad \text{(both the spectral norm of } \mathcal{T}_1 \text{ and } \widehat{\mathcal{U}}_r^1 \text{ are bounded by 1)} \tag{39}$$

$$= \left\|\mathcal{U}_r^1 - \widehat{\mathcal{U}}_r^1\right\|_{\bullet} + \left\|\mathcal{U}_r^2 \mathcal{T}_2 - \widehat{\mathcal{U}}_r^2 \widehat{\mathcal{T}}_2\right\|_{\bullet} \qquad \text{(reshaping does not change the norm)} \tag{40}$$

$$\leqslant \sum_{i=1}^{d-2} \left\|\mathcal{U}_r^i - \widehat{\mathcal{U}}_r^i\right\|_{\bullet} + \left\|\mathcal{A}_{d-1} - \widehat{\mathcal{A}}_{d-1}\right\|_{\bullet} \qquad \text{(by induction on } i\text{)} \tag{41}$$

Next, we derive perturbation bound for $\mathcal{U}_r^i$. Assume that all singular values of $\mathcal{A}_i$ have multiplicity 1, and then the perturbed version $\widehat{\mathcal{U}}_r^i$ due to sampling error can be parameterized as $\widehat{\mathcal{U}}_r^i = (\mathcal{U}_r^i + \mathcal{U}_\perp^i \mathcal{D})(\mathcal{I} + \mathcal{D}^\top \mathcal{D})^{-1/2}$ where $\mathcal{U}_r^{i\top} \mathcal{U}_\perp^i = 0$. Then

$$\left\|\mathcal{U}_r^i - \widehat{\mathcal{U}}_r^i\right\|_{\bullet} = \left\|\mathcal{U}_r^i - (\mathcal{U}_r^i + \mathcal{U}_\perp^i \mathcal{D})(\mathcal{I} + \mathcal{D}^\top \mathcal{D})^{-1/2}\right\|_{\bullet} \tag{42}$$

$$= \left\|(\mathcal{I} - (\mathcal{I} + \mathcal{D}^\top \mathcal{D})^{-1/2})\mathcal{U}_r^i - \mathcal{U}_\perp^i \mathcal{D}(\mathcal{I} + \mathcal{D}^\top \mathcal{D})^{-1/2}\right\|_{\bullet} \tag{43}$$

$$= \left\|(\mathcal{I} - (\mathcal{I} + \mathcal{D}^\top \mathcal{D})^{-1/2})\mathcal{U}_r^i\right\|_{\bullet} - \left\|\mathcal{U}_\perp^i \mathcal{D}(\mathcal{I} + \mathcal{D}^\top \mathcal{D})^{-1/2}\right\|_{\bullet} \tag{44}$$

$$\leqslant \text{higher order errors (will be dropped)} + \|\mathcal{D}\|_{\bullet} \tag{45}$$

$$\leqslant \frac{\left\|\mathcal{A}_i - \widehat{\mathcal{A}}_i\right\|_{\bullet}}{\lambda} \qquad \text{(Wedin's theorem)} \tag{46}$$

$$= \frac{\left\|\mathcal{P}_{i-1}\mathcal{C}_{X_{1:i};X_{i+1:d}} - \widehat{\mathcal{P}}_{i-1}\widehat{\mathcal{C}}_{X_{1:i};X_{i+1:d}}\right\|_{\bullet}}{\lambda} \tag{47}$$

$$(\mathcal{P}_{i-1} := ((\mathcal{U}_r^{i-1\top} \dots (\mathcal{U}_r^{2\top}(\mathcal{U}_r^{1\top} \otimes \mathcal{I}) \otimes \mathcal{I}) \dots \otimes \mathcal{I})) \tag{48}$$

$$\leqslant \frac{\left\|\mathcal{C}_{X_{1:i};X_{i+1:d}} - \bar{\mathcal{C}}_{X_{1:i};X_{i+1:d}}\right\|_{\bullet} + \sum_{j=1}^{i-1}\left\|\mathcal{U}_r^j - \widehat{\mathcal{U}}_r^j\right\|_{\bullet}}{\lambda} \tag{49}$$

$$\tag{50}$$

which is bounded in a recursive fashion. We now will derive a closed form bound for $\left\|\mathcal{U}_r^i - \widehat{\mathcal{U}}_r^i\right\|_{\bullet}$. For simplicity of notation, Let $a_i := \left\|\mathcal{U}_r^i - \widehat{\mathcal{U}}_r^i\right\|_{\bullet}$ and $\Delta := \left\|\mathcal{C}_{X_{1:d}} - \bar{\mathcal{C}}_{X_{1:d}}\right\|_{\bullet}$, we have that

$$\lambda\, a_1 \leqslant \Delta \tag{51}$$
$$\lambda\, a_2 \leqslant \Delta + a_1 \tag{52}$$
$$\lambda\, a_3 \leqslant \Delta + a_1 + a_2 \tag{53}$$
$$\lambda \dots \tag{54}$$
$$\lambda\, a_i \leqslant \Delta + \sum_{j=1}^{i-1} a_j \tag{55}$$

Rearranging terms, we have that

$$\begin{pmatrix} \lambda & 0 & 0 & \dots & 0 \\ -1 & \lambda & 0 & \dots & 0 \\ -1 & -1 & \lambda & \dots & 0 \\ \vdots & \vdots & \vdots & \ddots & \vdots \\ -1 & -1 & -1 & \dots & \lambda \end{pmatrix} \begin{pmatrix} a_1 \\ a_2 \\ a_3 \\ \vdots \\ a_i \end{pmatrix} \leqslant \begin{pmatrix} \Delta \\ \Delta \\ \Delta \\ \vdots \\ \Delta \end{pmatrix}. \tag{56}$$

Solving the above equation, we have that

$$a_i \leqslant \frac{(1+\lambda)^{i-1}}{\lambda^i}\Delta \qquad \Leftrightarrow \qquad \left\|\mathcal{U}_r^i - \widehat{\mathcal{U}}_r^i\right\|_\bullet \leqslant \frac{(1+\lambda)^{i-1}}{\lambda^i}\left\|\mathcal{C}_{X_{1:d}} - \bar{\mathcal{C}}_{X_{1:d}}\right\|_\bullet \qquad (57)$$

Therefore, equation (41) is bounded by

$$\left\|\widetilde{\mathcal{C}}_{X_{1:d}} - \widehat{\mathcal{C}}_{X_{1:d}}\right\|_\bullet = \sum_{i=1}^{d-2}\left\|\mathcal{U}_r^i - \widehat{\mathcal{U}}_r^i\right\|_\bullet + \left\|\mathcal{A}_{d-1} - \widehat{\mathcal{A}}_{d-1}\right\|_\bullet \qquad (58)$$

$$\leqslant \sum_{i=1}^{d-2}\frac{(1+\lambda)^{i-1}}{\lambda^i}\left\|\mathcal{C}_{X_{1:d}} - \bar{\mathcal{C}}_{X_{1:d}}\right\|_\bullet + \left\|\mathcal{C}_{X_{1:d}} - \bar{\mathcal{C}}_{X_{1:d}}\right\|_\bullet + \sum_{i=1}^{d-2}\frac{(1+\lambda)^{i-1}}{\lambda^i}\left\|\mathcal{C}_{X_{1:d}} - \bar{\mathcal{C}}_{X_{1:d}}\right\|_\bullet$$

$$(59)$$

$$\leqslant \frac{(1+\lambda)^{d-2}}{\lambda^{d-2}}\left\|\mathcal{C}_{X_{1:d}} - \bar{\mathcal{C}}_{X_{1:d}}\right\|_\bullet \qquad (60)$$

Furthermore, based on the concentration inequality for kernel embeddings [13], we have that with high probability at least $1-\delta$

$$\left\|\widetilde{\mathcal{C}}_{X_{1:d}} - \widehat{\mathcal{C}}_{X_{1:d}}\right\|_\bullet \leq \frac{(1+\lambda)^{d-2}}{\lambda^{d-2}}\left\|\mathcal{C}_{X_{1:d}} - \bar{\mathcal{C}}_{X_{1:d}}\right\|_\bullet \leqslant \frac{(1+\lambda)^{d-2}c}{\lambda^{d-2}\sqrt{n}} \qquad (61)$$

with some constant $c$ associated with the kernel and the probability $\delta$. ∎