[Reviews · NeurIPS 2013]

Submitted by Assigned_Reviewer_5

Summary of the paper: The RKHS embedding of a joint probability distribution between two variables involves the notion of covariance operators. For joint distributions over multiple variables, a tensor operator is needed. The paper defines these objects together with appropriate inner product, norms and reshaping operations on them. The paper then notes that in the presence of latent variables where the conditional dependence structure is a tree, these operators are low-rank when reshaped along the edges connecting latent variables. A low-rank decomposition of the embedding is then proposed that can be implemented on Gram matrices. Empirical results on density estimation tasks are impressive.

Because of notational overhead, I found this paper bit hard to read. Some simplification, wherever possible, would be helpful.

Is the assumption that X_1-toX_d all are on the same domain a simplification? In general, the tensor operators could be defined over multiple different kernels each acting on their own domain.

In Algorithm 1, will the sequence of reshaping operations matter? It is not clear to me whether the solution is unique and whether Algorithm 1 is solving a tractable low-rank approximation problem (tensor decompositions are usually non-convex heuristics).

Typos:

Eqn 11: p(x_1...x_3): x_3 should be x_d
First line of Section 4: kenrel -> kernel
Summary: Exploiting low-rank structure in RKHS embedding techniques is important. The approach seems to be correctly derived and the empirical results are excellent. The paper is overall well written though the notation could be simplified a bit.

Submitted by Assigned_Reviewer_6

The authors make a kernel extension of paper [1]. That is they give an svd based algorithm for low rank approximations of embeddings with an underlying latent structure. Weak guarantees are given under the assumption that low rank approximations of certain key objects have a low error. Experimental results are promising.

Section 4 is cryptic. This is Section 3.1 in [1] in which the ideas seem more clear due to the simpler setting. Eq 7 (and the follow ups) are the point of concern ( I'm actually not sure how they define C_X|Z; before sec 3 they write mu_X|z = C_X|Z phi(z) so C_X|Z : H -> F; but in the sentence above it is said that C_X|Z acts on z. I think for 7 the phi definition is wrong and they mean the second (?) let's go with this one). They want to condition to extract the conditioning variable and show a low rank of the covariance matrix.

I guess the argument is

1) condition with Z and average over Z.

2) Fubini theorem (?) for vector integrals(?) -- I guess technical details, e.g. when can we apply this for vector integrals, are not really of interest to the community and I'm happy to follow the heuristic argument of the authors

3) The next step is then really cryptic. I think what they do is to denote the vector (E_X1|Z=(1 0) [phi(X1)] E_X1|Z=(0 1) [phi(X1)] ) with E_X1|Z [phi(X1)] and then the multiple this with Z itself to regain the conditional expectation.

The notation really needs to be cleaned up a lot. Actually it might be better to first go through a short version of Sec 3.1 from [1] and just show what changes in the kernel setting.

4) The next step is then to substitute the cond exp vector with C_X1|Z . The whole thing is just super confusing since the notation is not explained and it is up to the reader to guess what the objects are.

#

I must admit that I found it hard to follow the algorithm discussion in Sec 5/6. E.g. in Sec 5 the svd of A_1 is understandable, but I guess ,this is the trivial part. The main algorithm seems to be then in the sentence: "This leads to the first intermediate tensor G_1 = U_r and we reshape S_r V_r and recursively decompose it" which I admit does not give me much of a picture of what is going on. As far as I get it they split (say for their example Fig 2 c) the problem into the first Z1- X1 pair and all the rest and perform an svd on the covariance between the first X1 and this potentially really long tensor.

I'm wondering here how this can lead to anything good. This sounds really like a formidable challenge if you have an averaged length HMM, say with d=100, to get anything of relevance from the svd??

#

The bounds in sec 7 are a bit odd. First, the statement "More specifically, we will bound, in terms of generalized Frobenius norm || ... ||, the difference between the true kernel embeddings and the low rank kernel embeddings estimated from a set of n iid sample {..} " is misleading, since they make the assumption that they have already low rank approximations of different covariance operators with error at max epsilon. So what they really seem to do here is just to say if the individual estimates are very good then the chained together is not so bad either. But obviously getting these guarantees on the low rank approximations will be a real difficulty.

The next thing with the bound is the 1/lambda^(d-2). lambda is the smallest singular value of a family of covariance matrices and d the dimension of the latent variable. I would expect that this lambda can get really small as you go over quite a number of matrices. The authors also argue in a similar line and say that lambda might be misleading and that in experiments the behaviour differs significantly from the exponential dependence of performance on lambda.

So this seems to say that under the rather strong conditions they found a bound that is far from tight.

Also, in the introduction they motivate with a mixture of Gaussians and argue that their estimate outperforms kernel density estimators. But there seems to be no statements to support this like asymptotic results or bounds.

#

In general there is significant overlap with the paper [1]. The authors say at one point that it is a kernel generalization of [1]. Some of the arguments look extremely similar like the low rank k thing in Sec 4 and Sec 3.1 in [1]. A clear discussion of what are the significant new findings compared to [1] must be given.

#

A spell-checker would also be much appreciated, so would be checking the math parts for typos and an overall polishing of the presentation.

#

The experimental results are the positive aspect of the paper where the method outperforms a number of competitors.

#

In summary I think that the paper has one main idea which is to kernelize the approach in [1]. The novelty itself does not feel ground breaking due to this. The paper is also lacking in presentation. I can't see people outside a small community to follow this paper through without significant difficulties. I think the main ideas could be nicely summarised in one or two paragraphs but it is currently a pain to extract these; the notation is guesswork and will frustrate a reader who is not from the field or wants to go into details. There is no theory to support the density estimation point they make and the covariance approximation bounds are also not super significant since they make strong assumptions and the bounds seem to be not very tight. Also the link to paper [1] needs to be pointed out clearly with a proper discussion. On the plus side is Table 1 with the experimental results which seem promising.

[1] Hierarchical Tensor Decomposition of Latent Tree Graphical Models. Le Song et al 2013 ICML.
Summary: A paper with good experimental results but weak theoretical support and a weak presentation which has the risk to make it only accessible to a small group of experts. In total this could have been --and possibly should have been -- a much nicer paper.

Submitted by Assigned_Reviewer_7

The authors present a robust low rank kernel embedding related to higher order tensors and
latent variable models. In general the work is interesting and promising. It provides synergies between machine learning, kernel methods, tensors and latent variable models.

- Introduction: "appear to be much closer to the ground truth". Though the intention of the authors is good to introduce the method in such a way, technically speaking it is not convincing.
The model selection should be done in an optimal way for each individual method in order to have a fair comparison. The same bandwidth is used now. This should in fact be optimized for Figure 2(b) and 2(c) independently.

On the other hand in Section 8 the authors have done a good effort for making fair comparisons between different methods.

- It would be good to explain more exactly what you mean by misspecification. It is not sufficiently clear whether this is a perturbation on the model of Fig.2 or a contamination
on the given data distribution. It would also be good to give a simple and concrete example
of this. Currently the notion of misspecification as explained in the introduction and other sections is too vague.

- Concerning performance guarantees it is not clear what the results mean in terms of estimating
the unknown function f. Some comments or discussion on this would be good if possible.

- Though a good effort has been done in comparing with other methods, it might be that these methods are too basic. If you could show that the proposed method would outperform the following more advanced methods I would be more convinced:

M. Girolami, Orthogonal series density estimation and the kernel eigenvalue problem, Neural Computation, 14(3), 669-688, 2002.

JooSeuk Kim, Clayton D. Scott; Robust Kernel Density Estimation, JMLR, 13(Sep):2529−2565, 2012.

- The work relates to tensors and kernels.

The following previous work also related SVDs, tensors and kernels:

Signoretto M., De Lathauwer L., Suykens J.A.K., A Kernel-based Framework to Tensorial Data Analysis, Neural Networks, 24(8), 861-874, 2011.

Signoretto M., Olivetti E., De Lathauwer L., Suykens J.A.K., Classification of multichannel signals with cumulant-based kernels, IEEE Transactions on Signal Processing, 60(5), 2304-2314, 2012.

- Throughout the paper it is not always clear whether a latent variable model should be given or not. Is the method also able to discover the structure or not?

I have read the author replies. I would like to thank the authors for the additional comments and clarifications.









Summary: Interesting connections between higher order tensors, kernels and latent variable models, but some parts can be improved and clarified.

Author Feedback

Author rebuttal: We thank the reviewers for their comments.
We will simplify the notation so that the paper will be more accessible to beginners as well as experts.

Reviewer 5:

The assumption that all variables have the same domain is just for simplicity of exposition. The method is applicable to the cases where the variables have different domains.

Reviewer 6:

The reviewer may have some misunderstanding here.
Equation (7) is correct. As we explained in the paper, for discrete hidden variable Z, we represent it as the standard basis in R^r. For instance, when r=3, Z can have three value (1, 0, 0), (0, 1, 0) and (0, 0, 1). In this case, linear kernel < Z, Z' > = Z^t Z is used. That is \phi(Z) = Z, and the notation in (7) is consistent with the conditional embedding operator. These is standard definition in kernel methods and kernel embeddings.

Furthermore, when Z is discrete, the conditional embedding operator reduces a separate embedding \mu_{X|z} for each conditional distribution P(X|z). Conceptually, we can concatenate these \mu_{X|z} for different value of z in columns (\mu_{X|(1,0,0)}, \mu_{X|(0,1,0)}, \mu_{X|(0,0,1)}), and use the standard basis representation of Z to pick up the corresponding embedding (or column). We derive it using Hilbert space embedding concepts which is strictly more general than those in paper [1].

Due to space constraints, the derivation details for the algorithm has been placed in appendix A (equation 21-23 and the paragraph around).

Note that our derivation is for the most general case where all conditional distribution (or conditional probability tables) are different. For HMMs with shared CPTs, one will use our algorithm to estimate the low rank embeddings of some repeated common structure, rather than blindly applying our algorithm to the entire HMM chain.

We used perturbation analysis to obtain a guarantee for low rank approximation of covariance operators. Similar techniques have been used for discrete cases (see for example Hsu et al. a spectral algorithm for hidden Markov models). One can obtain more sophisticated guarantee by generalizing the work of Negahban et all. estimation of (near) low-rank matrices with noise and high-dimensional scaling to integral operator (see Rosasco et al. on learning with integral operators). We emphasize that it is not a trivial task to combine low rank approximation guarantees from individual covariance operators to obtain a guarantee for the entire recursive decomposition algorithm.

Our results show that the recursive decomposition algorithm can lead to consistent estimator which is an important theoretical aspect of the method. Dependence in the bottom singular values are common in the perturbation analysis. We are not aware of any other guarantees for this challenging problem in the literature. To obtain the sharpest analysis for the rate of convergence will be a subject of future work.

Density estimation is just one application of our low rank embedding. The key advantage of the low rank embedding is that we can use it to evaluate integral of a function from the RKHS.

Reviewer 7:

In the example in introduction, we have chosen the best bandwidth for KDE via cross-validation. So the comparison is fair.

Model misspecification means that the supplied latent variable model structure can be incorrect.

We can compare the methods of Girolami and Kim et al. Note that both work do not consider low rank structure in the data, which is the key challenge we attack in our paper.

In Signoretto et al., each data point itself is a tensor which is quite different from our embedding setting here where we have a latent structure underlying the distribution.